

# No island-effect on glucocorticoid levels for a rodent from a near-shore archipelago

Nathan D. Stewart[1], Gabriela F. Mastromonaco[2] and Gary Burness[3]

[1] Environmental and Life Sciences Graduate Program, Trent University, Peterborough, Ontario, Canada
[2] Reproductive Physiology, Toronto Zoo, Toronto, Ontario, Canada
[3] Department of Biology, Trent University, Peterborough, Ontario, Canada

## ABSTRACT

Island rodents are often larger and live at higher population densities than their mainland counterparts, characteristics that have been referred to as "island syndrome". Island syndrome has been well studied, but few studies have tested for island-mainland differences in stress physiology. We evaluated island syndrome within the context of stress physiology of white-footed mice (*Peromyscus leucopus*) captured from 11 islands and five mainland sites in Thousand Islands National Park, Ontario, Canada. Stress physiology was evaluated by quantifying corticosterone (a stress biomarker), the primary glucocorticoid in mice, from hair and its related metabolites from fecal samples. White-footed mice captured in this near-shore archipelago did not display characteristics of island syndrome, nor differences in levels of hair corticosterone or fecal corticosterone metabolites compared with mainland mice. We suggest that island white-footed mice experience similar degrees of stress in the Thousand Islands compared with the mainland. Although we did not find evidence of island syndrome or differences in glucocorticoid levels, we identified relationships between internal (sex, body mass) and external (season) factors and our hormonal indices of stress in white-footed mice.

Corresponding author
Gary Burness, garyburness@trentu.ca

## INTRODUCTION

Studying island ecosystems and species has been central to the development of ecological and evolutionary theory (*Foster, 1964*; *MacArthur & Wilson, 1967*; *Van Valen, 1973*; *Lomolino et al., 2012*; *Warren et al., 2015*). Island communities tend to have low species diversity compared with mainland systems (*Losos & Ricklefs, 2009*), including fewer native predators (*Blackburn et al., 2004*). In response to decreased predator pressure and interspecific competition, combined with changes in food availability on islands, small mammals evolve towards gigantism upon arrival to islands while larger species often display dwarfing (*Lomolino et al., 2012*). This pattern, observed across numerous archipelagos, contributes to the evolutionary trend called the "island rule" (*Van Valen, 1973*). Behavioural and morphological changes co-occur in small mammals following island

colonization, with individuals becoming less aggressive to conspecifics and demonstrating less predator avoidance behaviour (*Adler & Levins, 1994*).

The combination of increased body size with changes in behaviour and demography has been referred to as "island syndrome" in rodents (*Adler & Levins, 1994*). Although morphological and behavioural traits associated with island syndrome have been relatively well-characterized, few studies have focused on the effect of island life on stress physiology (but see *Clinchy et al., 2004*; *Müller et al., 2007*).

When an animal encounters a perceived stressor, its hypothalamic-pituitary-adrenal (HPA) axis is activated, resulting in increased secretion of glucocorticoid hormones (GCs; *Sapolsky, Romero & Munck, 2000*). Multiple environmental factors can influence GC levels, including predation (*Clinchy et al., 2011*; *Sheriff, Krebs & Boonstra, 2011*), food availability (*Kitaysky, Wingfield & Piatt, 1999*; *Walker, Wingfield & Boersma, 2005*) and population density (*Boonstra & Boag, 1992*; *Harper & Austad, 2004*; *Dettmer et al., 2014*; *Blondel et al., 2016*). Elevated GC levels are involved in preparation for future stressors by shifting resources from reproduction and digestion toward replenishing energy stores used during the initial stress response (*Romero & Wingfield, 2016*). Although short-term elevations of GCs are presumed to be adaptive, chronically elevated GCs may correlate negatively with indices of wildlife health (*Stothart et al., 2016*), condition (*Injaian et al., 2019*), and survival (*Wilkening & Ray, 2016*).

In wildlife studies, stress has traditionally been evaluated by quantifying circulating GC levels from blood samples (*Sapolsky, 1982*; *Wingfield, Smith & Farner, 1982*). However, there is increasing interest in quantifying GCs from less-invasive alternative sources, including saliva, feces and hair (*Sheriff et al., 2011*). Importantly, comparing measures of GC levels in different materials allows for stress to be evaluated over different time scales (*Sheriff et al., 2011*). For example, wild eastern chipmunks (*Tamias striatus*) from more open environments had higher fecal cortisol metabolites than those from more forested areas; however, there was no difference in long term stress between the two groups based on hair cortisol (*Mastromonaco et al., 2014*).

Although concerns have been raised regarding interpretation of both fecal (*Goymann, 2012*) and hair GCs (*Sharpley, Kauter & McFarlane, 2009*; *Keckeis et al., 2012*; *Stewart et al., 2018*), a meta-analysis suggests they are useful metrics for quantifying an individual's response to environmental stressors (*Dantzer et al., 2014*).

Hair GC levels are influenced by both internal and external factors (*Romero & Wingfield, 2016*; *Heimbürge, Kanitz & Otten, 2019*). Hair GC levels vary within individuals between body regions (*Macbeth et al., 2010*; *Acker, Mastromonaco & Schulte-Hostedde, 2018*) and among individuals by age (*Dettmer et al., 2014*), body size (*Waterhouse et al., 2017*), condition (*Cattet et al., 2014*), sex (*Stewart et al., 2018*), food availability (*Cattet et al., 2014*) and season (*Martin & Réale, 2008*). Further, a study of captive rhesus macaques showed that hair GC levels increased with population density (*Dettmer et al., 2014*). Fecal GC levels are also influenced by internal and external factors in wild mammals (*Hayssen, Harper & DeFina, 2002*; *Smith et al., 2012*; *Mastromonaco et al., 2014*).

Animals display a range of physiological adaptations to island life. For example, immunological differences exist between bluebirds (*Sialia sialis*) from Bermuda and the

continental United States (*Matson et al., 2014*). Caviomorph rodents from small islands have lower basal metabolic rates than their mainland counterparts (*Arens & McNab, 2001*). With respect to stress physiology, circulating corticosterone levels were lower in blue tits (*Parus caeruleus*) from the island of Corsica than mainland France (*Müller et al., 2007*). Although some data are available for changes in glucocorticoid levels in response to island life for birds, demonstration of a pattern similar to the island rule or island syndrome is lacking. The ecological factors shown to affect GC levels in wildlife (e.g., predation, competition, and resource availability) are also thought to account for island syndrome in rodents (see *Adler & Levins, 1994*). Given that island syndrome is largely attributed to decreased predation pressure experienced by island rodents (*Adler & Levins, 1994*), and that low predation is associated with low GC levels (*Clinchy et al., 2011*), we predicted that populations of island rodents displaying island syndrome would also have lower GC levels than their mainland relatives.

Although many island-mainland comparisons of behavioural and physiological characteristics have been made using wildlife from isolated oceanic islands (*Müller et al., 2007*; *Novosolov, Raia & Meiri, 2013*; *Matson et al., 2014*; *Cuthbert et al., 2016*), evidence of island syndrome in rodents has been found in relatively near-shore archipelagos in both freshwater and marine environments (*Lomolino, 1984*; *Adler & Tamarin, 1984*). The Thousand Islands are an archipelago in the St. Lawrence River, whose islands are divided between Canada and the United States. The Thousand Islands region has served as the site for numerous studies of island biogeography in small mammals (*Lomolino, 1982*; *Lomolino, 1984*; *Werden et al., 2014*). Mammalian species richness in the Thousand Islands is positively related to island area (*Lomolino, 1982*), and the body size of meadow voles (*Microtus pennsylvanicus*) and short-tailed shrews (*Blarina brevicauda*) increases with the degree of isolation from the mainland (*Lomolino, 1984*). On the basis of these trends in species richness and morphology, we selected a portion of the islands belonging to Thousand Islands National Park in Ontario, Canada as our study site. As a study species, we focused on white-footed mice (*Peromyscus leucopus*) because they are the most abundant small mammal in Thousand Islands National Park (*Werden et al., 2014*), and *Peromyscus* has been the focus of many studies that provide the basis for island syndrome (*Adler & Levins, 1994*).

The current study had two aims: 1. to test if white-footed mice in the Thousand Islands display characteristics of island syndrome, including greater body mass and higher relative abundance than their mainland counterparts, and 2. to test if there were differences in stress physiology between island and mainland mice. We tested these aims under the hypothesis that island syndrome includes changes in stress physiology. To test our hypothesis, we compared corticosterone (the major GC in mice and rats; *Keeney, Jenkins & Waterman, 1992*) in hair ($CORT_{hair}$), and its metabolites in feces ($CORT_{feces}$), of white-footed mice captured at multiple island and mainland locations in a near-shore archipelago; from sites in Thousand Islands National Park in Ontario, Canada. We predicted that if white-footed mice in the Thousand Islands were more abundant and larger than mainland mice (as expected by "island syndrome"), then island mice would also have lower $CORT_{hair}$ and $CORT_{feces}$ levels. Because aspects of island syndrome in rodents are affected by island area

and distance from the mainland (*Adler & Levins, 1994*), we also predicted that white-footed mice would have lower GC levels on the smaller and more isolated islands in the archipelago. Finally, we evaluated how body mass affected CORT in white-footed mice, and whether CORT varied between seasons and with population density.

## MATERIALS & METHODS

The Trent University Animal Care Committee (protocol numbers 23877 and 24341) approved all procedures prior to working with the animals. Trapping in the Thousand Islands National Park was approved via a Parks Canada Research and Collection Permit (No. 22959).

### Study species and location

The white-footed mouse is a small, nocturnal rodent that inhabits deciduous and mixed forests in the eastern United States and southern edge of Canada (*Werden et al., 2014*). White-footed mice in Ontario breed from April–August, and females have litters of approximately five individuals on average (*Millar, Wille & Iverson, 1979*). Female white-footed mice in southern Ontario produce 1–4 litters per year (*Harland, Blancher & Millar, 1979*). Moulting in *Peromyscus* precedes or follows energetically costly periods such as reproduction, and begins in March in white-footed mice in the nearby state of New York (*Pierce & Vogt, 1993*). Our small mammal trapping period coincided with the breeding season, when most individuals would be displaying their summer pelage.

All trapping locations were located in Thousand Islands National Park in Ontario, Canada (Fig. 1). Selection of trapping locations was limited to locations within Thousand Islands National Park, and those islands that were large enough to place trapping grids. These islands are also visited by visitors to the National Park, so traps had to be placed away from walking paths and campsites. We trapped on 11 islands and at five mainland sites during two years (2015 and 2016). Mainland sites were located within 2 km of the St. Lawrence River (Fig. 1). We targeted wooded areas as opposed to open fields so that relative abundance could be compared between locations and to increase trapping success. Island area and distance from the mainland were calculated using ArcMap (Version 10.4.1; see Table S1).

### Small mammal trapping

We trapped during three periods: summer 2015 (July–August), spring 2016 (May-June) and summer 2016 (July–August). Efforts were made to alternate between trapping on islands and mainland sites; however, weather conditions occasionally dictated access to island sites (trapping dates are provided in Table S1). Sherman live-traps (H.B. Sherman Traps, Inc.,Tallahassee, FL, USA) were set 10 m apart in rectangular grids of varying size. The majority of grids were arranged in a 7 by 7 formation (49 traps in total), however some areas on small islands were limited by pedestrian paths, which necessitated using smaller grids (5 by 5), or in one case, transects (Mermaid Island). Hulled sunflower seeds were used as bait and natural cotton bedding was provided for warmth. Traps were set in the evening (ca. 1800 h) and checked in the morning (ca. 0700 h) to target the active period

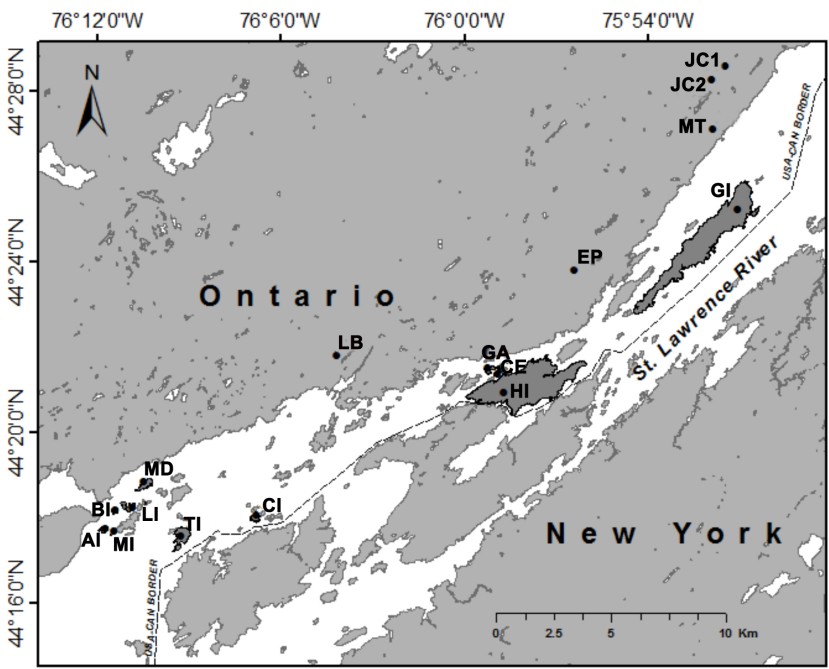

**Figure 1** **Trapping locations of white-footed mice (*Peromyscus leucopus*) in Thousand Islands National Park, Ontario, Canada (*USGS, 2016*).** Islands on which trapping occurred are shaded in dark grey. Abbreviations: AI, Aubrey Island; BI, Beau Rivage Island; CE, Constance Island; CI, Camelot Island; EP, Escot Property; GA, Georgina Island; GI, Grenadier Island; HI, Hill Island, JC1, Jones Creek 1; JC2, Jones Creek 2; LB, Landon Bay; LI, Lindsay Island; MD, McDonald Island; MI, Mermaid Island; MT, Mallorytown; and TI, Thwartway Island.

of white-footed mice. Trapping periods generally consisted of 2–4 nights of consecutive trapping.

Upon capture, white-footed mice were weighed (±1 g) and a patch of hair (ca. 1 × 1 cm) was shaved from the rump of each individual, above the right-hind limb using an electric razor (Remington[TM] Model PG6025), collecting the entire length of each shaft from the skin to the distal end of the shaft. We standardized the shaving location because hair GC levels can vary among body regions (*Macbeth et al., 2010*; *Acker, Mastromonaco & Schulte-Hostedde, 2018*). Each white-footed mouse was ear-tagged to recognize recaptured individuals, and then released. The razor blades were cleaned with alcohol swabs between shaving each animal. Hair samples were stored in Fisherbrand[TM] Snap-Cap[TM] Flat-Top Microcentrifuge Tubes in the dark at ambient temperature (approx. 22 °C) until hair hormone analysis (2–5 months later). Coat colour and body mass were used to exclude juveniles from our analyses. Coat colour and stage were occasionally noted (grey, brown, reddish-brown or moulting) in 2015, and always noted in 2016. We identified juveniles by their grey pelage and in the absence of coat colour data, individuals ≤14 g were excluded (*Wolff, 1984*).

White-footed mouse feces were collected from traps using forceps. Fecal samples were stored in Fisherbrand[TM] Snap-Cap[TM] Flat-Top Microcentrifuge Tubes and placed in a

cooler with ice packs until they could be stored in a liquid nitrogen-cooled dry-shipper (within 6 h of collection). Fecal glucocorticoid metabolites are stable over the timeframe of our sample collection. For example, fecal samples maintained for 5-hours at room temperatures had similar levels as samples immediately frozen following collection (*Dantzer et al., 2010*). At ambient temperature, metabolites are reported to be stable for up to 48-h (*Parnell et al., 2015*). Soiled traps were cleaned with 70% ethanol between uses to ensure that the feces collected from each trap belonged to the animal caught in the trap that night. Samples were then transferred to a −80C freezer until hormone extraction (2–9 months in freezer). At sub-zero temperatures, fecal glucocorticoid metabolites are stable for at least a year (*Beehner & Whitten, 2004*).

## Relative Abundance

As a proxy for the population density, we calculated relative abundance of white-footed mice at each site during each of the 3 trapping periods. Relative abundance was calculated as catch-per-unit-effort (CPUE), presented in number of white-footed mice captured per hundred trap-nights. We corrected for tripped traps following the correction factor equation (*Nelson & Clark, 1973*):

$$CPUE = A \times 100/(TU - S/2)$$
$$TU = P \times N$$

where A = number of white-footed mice caught; TU = trapping units, calculated as total number of trap nights per site session; P = number of nights in each trapping session; N = number of traps set each night; and S = total sprung traps. This approach has been widely used in other studies of small mammal ecology (*Parker et al., 2016*; *Gill et al., 2018*; *Fauteux et al., 2018*).

## Hormone extraction and analysis

The use of fecal CORT metabolites and hair CORT have been validated as measures of stress in laboratory mice (*Mus musculus*; *Daniszová et al., 2017*; *Erickson, Browne & Lucki, 2017*). Hair corticosterone ($CORT_{hair}$) and fecal corticosterone metabolites ($CORT_{feces}$) were extracted with methanol (100% for hair, 80% for feces) following *Mastromonaco et al. (2014)* and *Stewart et al. (2018)*. Hair samples were cut into 5 mm pieces and weighed into 7 ml glass scintillation vials. The samples were washed with 100% methanol by vortexing for 10 s followed by immediate removal of the methanol. Immediately thereafter, 100% methanol was added to the samples at a ratio of 0.005 g/ml. Samples were vortexed for 10 s and mixed for 24 h on a plate shaker. After 24 h, the vials were centrifuged for 10 min at 2,400 g and the supernatants were transferred into clean glass vials. Fecal samples were thawed and weighed into 7 ml glass scintillation vials to which 80% methanol in water (v:v) was added at a ratio of 0.05 g/ml. The samples were vortexed for 10 s and mixed overnight on a plate shaker. The vials were then centrifuged for 10 min at 2,400 g and the supernatants were transferred into clean glass vials. The supernatants from extracted hair and fecal samples were stored sealed at −20 °C for 1–9 months until they were evaporated and analyzed. Dried-down hair extracts (600 µl per sample) were reconstituted in 150 µl

EIA buffer (0.1 mM sodium phosphate buffer, pH 7.0, containing 9 g of NaCl and 1 g of bovine serum albumin per litre) resulting in a 4-fold concentration. Dried-down fecal extracts (200 µl per sample) were reconstituted in 200 µl EIA buffer and diluted for a final 1:20 dilution.

To quantify $CORT_{hair}$ and $CORT_{feces}$, we used an enzyme immunoassay (EIA) following methods described by *Baxter-Gilbert et al. (2014)*. Microtitre plates were coated with 0.25 µg/well goat anti-rabbit IgG polyclonal antibody (Sigma-Aldrich, Mississauga, ON, Canada; 1:200,000 in coating buffer, 50 mM bicarbonate buffer, pH 9.6) and incubated overnight at room temperature. Plates were washed with 0.05% Tween 20, 0.15 M NaCl solution and blocked with 250 µl EIA buffer for 1 hr at room temperature. Plates were then loaded with 50 µl corticosterone standard (Steraloids Q1550; 39–10,000 pg/ml), reconstituted extracts and controls, followed by 100 µl horseradish peroxidase conjugate (1:1,000,000) and 100 µl corticosterone antiserum (1:200,000) (antibody lot: CJM006; C. Munro, University of California, Davis, CA, USA), all diluted in EIA buffer. Plates were incubated overnight at room temperature, and then washed and loaded with 200 µl of substrate solution (0.5 ml of 4 mg/ml tetramethylbenzidine in dimethylsulphoxide and 0.1 ml of 0.176 M $H_2O_2$ diluted in 22 ml of 0.01 M sodium acetate trihydrate [$C_2H_3NaO_2 \cdot 3H_2O$], pH 5.0). After 30 min incubation, colour reaction was stopped with 50 µl $H_2SO_4$ (1.8M) and absorbance was measured at 450 nm using a spectrophotometer (MRX$^e$ microplate reader, Dynex Technologies, Chantilly, VA). The assay cross-reactivities are: corticosterone (100%), desoxycorticosterone (14.25%), and other GC metabolites (<3%) (*Watson et al., 2013*). Inter- and intra-assay CV's were 13.9% and 4.4%, respectively, with 25 plates being run in total. Samples were run as duplicates, and only samples with <10% CV were accepted (if CV >10%, the sample was re-run). Because CORT is the dominant GC in white-footed mice, but is highly metabolized prior to excretion (*Touma et al., 2003*), we refer to the values obtained from fecal analysis by EIA as fecal CORT metabolites. All hormone concentrations are described as ng of CORT/g of feces or hair. Serial dilutions of pooled hair and fecal extract showed parallel displacement with the corticosterone standard curve (hair: $r = 0.997$, $p < 0.01$; fecal: $r = 0.996$, $p < 0.01$; Fig. S1). The recovery of known concentrations of corticosterone from mouse hair and fecal extracts were $102.7 \pm 4.3\%$ and $83.5 \pm 5.4\%$, respectively. The measured hormone concentrations in the spiked samples correlated with the expected concentrations (hair $r = 0.999$, $p < 0.01$; fecal $r = 0.998$, $p < 0.01$).

## Statistical analysis

We tested all of our predictions using linear-mixed effects models, with sampling site (specific island or mainland location) as a random effect that was nested within habitat type (whether the site was on an island or on the mainland) for island-mainland comparisons. Our analyses can be broken down into three groups: island-mainland comparisons, among-island comparisons, and seasonal analyses. With the exception of our analysis of relative abundance, island-mainland comparisons were made using only summer data (July–August captures) for 2015 and 2016 (spring 2016 data were excluded). For among-island comparisons, only island capture data from summer months were used and island size and

isolation were added as covariates. For seasonal analyses, only 2016 data were used because we had both spring and summer data for that year. Habitat type (island-mainland) and island size and isolation were dropped from testing for seasonal effects on morphological and physiological parameters. All corticosterone and body mass data were ln-transformed, and island area and distance to the mainland were $\log_{10}$ transformed in all analyses to improve the normality of model residuals. Any visibly pregnant females ($n = 15$; *Millar, Wille & Iverson, 1979*) were excluded from all analyses (*Novikov & Moshkin, 1998*; *Young et al., 2006*), with the exception of relative abundance. If an individual was captured more than once, only data from the first capture were used.

In all analyses, two-way interactions were included in initial models for those variables that were expected to be biologically relevant (sex and season) and pertinent to our hypotheses (habitat type and island measurements). Sampling site was retained as a random effect in all models and nested within habitat type for island-mainland comparisons. The full model for the island-mainland comparison of relative abundance included habitat type, year and season with all two-way interactions. The full model for comparing relative abundance among islands included year and season with their interaction, and covariates for island size and isolation with their interaction. The full model for the island-mainland comparison of body mass included habitat type, sex and year with all two-way interactions. The full model for comparing body mass among islands included sex and year with their interaction, and covariates for island size and isolation with their interaction. The full models for the island-mainland comparisons of both $CORT_{hair}$ and $CORT_{feces}$ included habitat type, sex and year with all two-way interactions, and covariates for relative abundance and body mass. The full models for comparing $CORT_{hair}$ and $CORT_{feces}$ among islands included sex and year with their interaction, covariates for body mass, relative abundance, and island size and isolation, and the interaction between island size and isolation. The full model for comparing $CORT_{hair}$ and $CORT_{feces}$ between seasons included sex and season with their interaction, and body mass as a covariate.

To reduce model complexity, non-significant two-way interactions ($p > 0.05$) were dropped to test the variables of greatest interest (habitat type, sex, and season). Two-way interactions were dropped in order of largest $p$-value, and then remaining fixed effects were also dropped in that order. Habitat type, sex, year and island size and distance to the mainland (for among island analyses), were always retained, with the exception of testing seasonal variation. Full models (not including two-way interactions) were presented if model reduction did not result in significance of main effects.

Analyses were conducted using RStudio (Version 0.99.484, RStudio, Inc). Linear-mixed effects models were fit with the lmer function of the "lme4" package (Version 1.1.7) using restricted maximum likelihood (REML) and non-standardized variables. Results, including $p$-values, $t$-values, and Satterthwaite approximations to degrees of freedom, were obtained using the "summary" function of the "lmerTest" package (Version 2.0.20; *Kuznetsova, Brockhoff & Christensen, 2016*). Goodness of fit was assessed using marginal$_{(M)}$ and conditional$_{(C)}$ pseudo $R^2$ values ($R^2_{GLMM}$; *Nakagawa & Schielzeth, 2013*) calculated with the "r.squaredGLMM" function in the "MuMIn" package (Version 1.13.4; *Bartoń, 2016*). $R^2_{GLMM(M)}$ represents the proportion of the variation explained by the fixed effects

alone, and $R^2_{\text{GLMM(C)}}$ represents the proportion of variation explained by both the fixed and random effects (*Nakagawa & Schielzeth, 2013*). Correlations were tested using the "cor.test" function in R, which calculates a *p*-value based on Fisher's Z transformation.

## RESULTS

We caught 408 individual white-footed mice during 2015-2016; 17 individuals were recaptured between trapping periods. Trapping success was highly variable across sampling sites, and there were no consistent trapping patterns between mainland and island sites. For example, based on overall CPUE, trapping was more successful on some islands than on the mainland (as would be predicted via the "island rule"); however, on some islands there were zero captures during some trapping periods while mainland sites always yielded captures (Table S2). More traps were tripped at island sites (mean ± SD; 38% ± 10.8%) than mainland sites (23% ± 12.8%), and it was likely that trap disturbance contributed to the high degree of variation in trapping success across habitat types. Although trapping effort varied among sites due to some sites only being trapped during one sampling period, trapping effort was approximately equal between the two habitat types. Mainland sites received an average of 288 ± 151.1 trap nights across the entire sampling program, and island sites received an average of 264 ± 147.6 trap nights (Table S1).

### Relative abundance of mice did not differ between islands and the mainland

The final model for the island-mainland comparison of relative abundance of white-footed mice was reduced from the full model by removing two-way interactions. Contrary to expectations of the island rule, there was no difference in relative abundance of island and mainland white-footed mice ($p = 0.667$, Table 1). Relative abundance differed between years; it was 44% higher in summer 2015 than in summer 2016 ($p = 0.012$; Table 1). In 2016, in which we had data for both spring and summer, relative abundance was 74% higher in the summer than the spring ($p = 0.033$; Table 1). The final model for the among-islands comparison of relative abundance of white-footed mice was reduced from the full model by removing two-way interactions. CPUE decreased on all islands between 2015 and 2016 ($p = 0.011$). There was no effect of island area ($p = 0.603$) nor distance from the mainland ($p = 0.442$, Table 1) on relative abundance of white-footed mice.

Summer abundance of white-footed mice from individual sampling sites was positively correlated between years (11 sites, $r = 0.852$, $t_9 = 4.87$, $p = 0.0009$; Fig. 2). Although there was a general decrease in relative abundance between years across habitat types, CPUE was nearly equal between years for the three mainland sites that were trapped during both summers (Fig. 2). This is demonstrated in Fig. 2, where CPUE values for mainland sites fall close to the dotted line which represents the relationship if abundances were equal between years. All resampled island sites fall below that line, showing a decrease in abundance between years for island mice (Fig. 2).

### Body mass did not differ between island and mainland mice

The final model for the island-mainland comparison of body mass of white-footed mice was reduced from the full model by removing two-way interactions. Contrary to expectations

**Table 1 Factors predicting relative abundance of white-footed mice captured over two years, for two separate models (island- mainland comparison, and among-islands comparison).** Linear-mixed effects models were used for analysis (random effect: sampling site, nested within habitat type) of variation between island and mainland sites, and among islands in response to geographic variables. Marginal (M) and conditional (C) pseudo $R^2$ ($R^2_{\text{GLMM}}$) values are provided. Models were reduced by removing all non-significant two-way interactions.

| Dataset | Fixed effects | $\beta$ | se | df | $t$ | $p$ | $R^2_{\text{GLMM}}$ (M), (C) |
|---|---|---|---|---|---|---|---|
| Island and mainland mice | Intercept | 16,640 | 4.80 | 34 | 3.47 | 0.001 | 0.23, 0.68 |
| | Habitat (mainland) | 2.60 | 5.90 | 13 | 0.44 | 0.667 | |
| | Year (2016) | −8.88 | 3.27 | 22 | −2.72 | **0.012** | |
| | Season (summer) | 7.16 | 3.15 | 22 | 2.27 | **0.033** | |
| Island mice | Intercept | 40.00 | 32.05 | 8 | 1.25 | 0.244 | 0.27, 0.76 |
| | Year (2016) | −11.86 | 4.08 | 14 | −2.91 | **0.011** | |
| | Season (summer) | 7.48 | 4.05 | 14 | 1.85 | 0.085 | |
| | $\text{Log}_{10}$ (Area) | 2.72 | 5.03 | 7 | 0.54 | 0.603 | |
| | $\text{Log}_{10}$ (Distance) | −8.86 | 11.00 | 8 | −0.81 | 0.442 | |

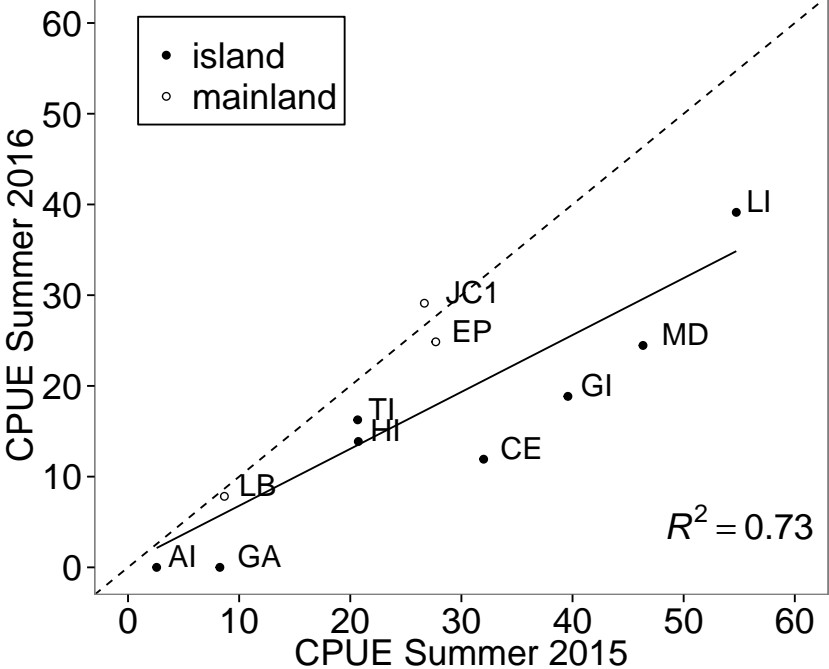

**Figure 2 Correlation between years in abundance of white-footed mouse (*Peromyscus leucopus*) at multiple trapping locations.** All trapping occurred in Thousand Islands National Park, Canada. The solid black line represents the linear relationship in abundance between the two years and the dashed line represents the predicted line if abundances were equal between years. Points that fall on the dotted line represent sites where abundance was nearly equal for white-footed mice between the two years. Abundance was measured in catch-per-unit-effort (CPUE, captures per 100 trap nights), which was corrected for tripped traps. Abbreviations represent individual sampling locations and correspond with those in Fig. 1.

**Table 2 Factors predicting body mass of white-footed mice during summer (July-August) in two years, for two separate models (island-mainland comparison, and among-islands comparison).** Linear-mixed effects models were used for analysis (random effect: sampling site, nested within habitat type). Marginal (M) and conditional (C) pseudo $R^2$ ($R^2_{GLMM}$) values are provided. Models were reduced by removing all non-significant two-way interactions.

| Dataset | Fixed effects | $\beta$ | se | df | $t$ | $p$ | $R^2_{GLMM}$ (M), (C) |
|---|---|---|---|---|---|---|---|
| Island and mainland mice | Intercept | 2.985 | 0.02 | 17 | 133.04 | <0.0001 | 0.02, 0.04 |
| | Habitat (Mainland) | −0.008 | 0.03 | 9 | −0.25 | 0.804 | |
| | Sex (Male) | 0.052 | 0.02 | 295 | 2.34 | **0.020** | |
| | Year (2016) | −0.004 | 0.02 | 292 | −0.16 | 0.875 | |
| Island mice | Intercept | 2.826 | 0.14 | 8 | 19.6 | <0.0001 | 0.04, 0.04 |
| | Sex (Male) | 0.061 | 0.03 | 203 | 2.37 | **0.019** | |
| | $Log_{10}$ (Area) | 0.014 | 0.02 | 5 | 0.76 | 0.476 | |
| | $Log_{10}$ (Distance) | 0.048 | 0.05 | 8 | 0.91 | 0.385 | |
| | Year (2016) | −0.017 | 0.03 | 203 | −0.61 | 0.545 | |

of the island rule, white footed mice from the Thousand Islands ($n = 209$) were not heavier than on the mainland ($n = 92$; $p = 0.804$; Table 2; Fig. 3), nor did body mass differ between years ($p = 0.875$). Male mice ($n = 181$; mean ± SD; 21.0 ± 3.7 g) were significantly heavier than females ($n = 120$; 20.2 ± 4.4 g) across habitat types ($p = 0.020$); however, the average difference was less than 1 g, and the model had little explanatory power ($R^2_{GLMM(M)} = 0.02$; $R^2_{GLMM(C)} = 0.04$; Table 2).

The final model for the among-islands comparison of body mass of white-footed mice was reduced from the full model by removing two-way interactions. For island mice, there was no effect of island area ($p = 0.476$, Table 2) nor distance from the mainland ($p = 0.385$) on body mass. Again, males were heavier than females on islands ($p = 0.019$); however, this effect explained little variation overall ($R^2_{GLMM(M)} = 0.04$; $R^2_{GLMM(C)} = 0.04$; Table 2).

## Hair corticosterone did not differ between island and mainland mice, but increased with body mass

The final model for the island-mainland comparison of $CORT_{hair}$ of white-footed mice was reduced from the full model by removing two-way interactions. There was no difference between $CORT_{hair}$ of island ($n = 188$) and mainland ($n = 82$) white-footed mice ($p = 0.4080$; Table 3, Fig. 3). Sex ($p = 0.9611$), year of capture ($p = 0.5159$) and relative abundance ($p = 0.3837$) also had no effect on $CORT_{hair}$ (Table 3). However, $CORT_{hair}$ increased with body mass ($p = 0.0011$; Table 3; Fig. 4). There was a small effect size for this model ($R^2_{GLMM(M)} = 0.05$, $R^2_{GLMM(C)} = 0.14$; Table 3).

The final model for the among-islands comparison $CORT_{hair}$ of white-footed mice was reduced from the full model by removing two-way interactions. Focusing only on island mice, neither island area ($p = 0.4807$, Table 3) nor distance from the mainland ($p = 0.6092$) predicted $CORT_{hair}$ levels. However, body mass continued to be a significant predictor of $CORT_{hair}$ for the model testing of only island mice ($p = 0.0031$; Table 3).

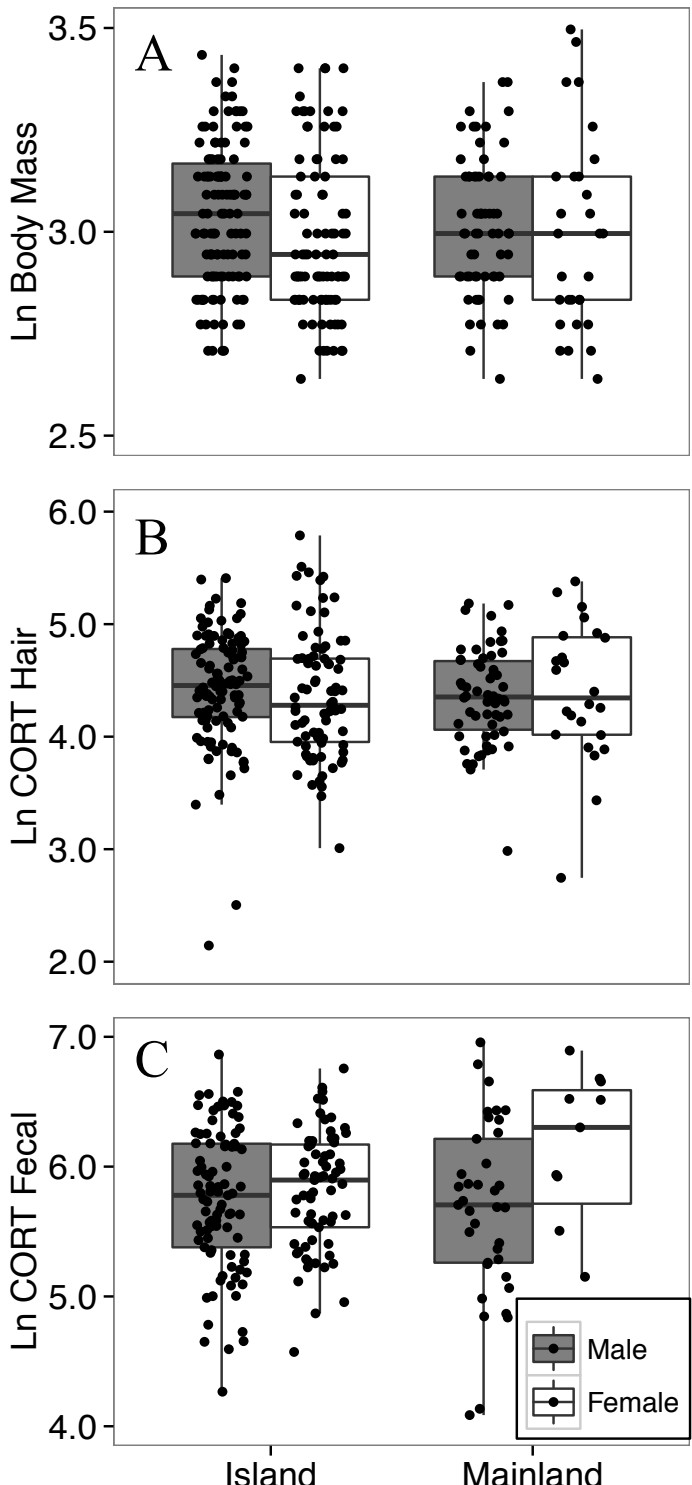

**Figure 3** Body mass (A), hair corticosterone (B), and fecal corticosterone metabolites (C) of white-footed mice captured during the summer (July–August) in two consecutive years. Body mass was measured in grams, and both measures of CORT were measured in ng/g. All response variables were ln-transformed.

**Table 3** Factors predicting hair corticosterone of white-footed mice during summer (July-August) in two years, for two separate models (island- mainland comparison, and among-islands comparison). Linear-mixed effects models were used for analysis (random effect: sampling site, nested within habitat type). Marginal (M) and conditional (C) pseudo $R^2$ ($R^2_{GLMM}$) values are provided. Models were reduced by removing all non-significant two-way interactions.

| Dataset | Fixed effects | $\beta$ | se | df | t | p | $R^2_{GLMM}$ (M), (C) |
|---|---|---|---|---|---|---|---|
| Island and mainland mice | Intercept | 2.812 | 0.539 | 253 | 5.22 | <0.0001 | 0.05, 0.14 |
| | Habitat (mainland) | −0.104 | 0.120 | 10 | −0.86 | 0.4080 | |
| | CPUE (Corrected) | −0.004 | 0.004 | 20 | −0.89 | 0.3837 | |
| | Sex (Male) | −0.003 | 0.064 | 261 | −0.05 | 0.9611 | |
| | Year (2016) | 0.051 | 0.078 | 104 | 0.65 | 0.5159 | |
| | Ln Body Mass | 0.568 | 0.172 | 261 | 3.31 | **0.0011** | |
| Island mice | Intercept | 2.824 | 0.923 | 15 | 3.06 | 0.0075 | 0.07, 0.16 |
| | Sex (Male) | −0.003 | 0.077 | 179 | −0.04 | 0.9713 | |
| | CPUE (Corrected) | −0.002 | 0.005 | 6 | −0.45 | 0.6675 | |
| | $\log_{10}$ Isl. Area | 0.076 | 0.097 | 3 | 0.79 | 0.4807 | |
| | $\log_{10}$ Isl. Distance | −0.134 | 0.243 | 4 | −0.55 | 0.6092 | |
| | Year (2016) | 0.077 | 0.110 | 19 | 0.70 | 0.4934 | |
| | Ln Body Mass | 0.631 | 0.211 | 179 | 3.00 | **0.0031** | |

## Fecal corticosterone metabolites did not differ between island and mainland mice

The final model for the island-mainland comparison of $CORT_{feces}$ of white-footed mice was reduced from the full model by removing two-way interactions, relative abundance and ln-transformed body mass, each of which had no significant effect. There was no difference between $CORT_{feces}$ of island ($n = 160$) and mainland ($n = 49$) white-footed mice during the summer months ($p = 0.8578$; Table 4). Male mice had lower $CORT_{feces}$ than females ($p = 0.0492$; Table 4; Fig. 3) and $CORT_{feces}$ was higher in the summer of 2016 than summer 2015 ($p < 0.0001$; Table 4).

The final model for the among-islands comparison $CORT_{feces}$ of white-footed mice was reduced from the full model by removing two-way interactions, relative abundance and ln-transformed body mass, each of which had no significant effect. There was no effect of either island area ($p = 0.9440$, Table 4) nor distance from the mainland ($p = 0.6320$) on $CORT_{feces}$ (Table 4). $CORT_{feces}$ levels were higher in 2016 than in 2015 ($p < 0.0001$), but unlike the model for the island-mainland comparison, there was no difference between sexes for $CORT_{feces}$ ($p = 0.1320$; Table 4).

## Hair corticosterone levels were lower in spring than summer for females, but not for males

The full model was retained for comparing $CORT_{hair}$ of white-footed mice between seasons. For $CORT_{hair}$ of white-footed mice collected in 2016 ($n = 147$), when we had data from both spring and summer, there was a significant interaction between sex and season ($p = 0.0009$, Table 5). Female $CORT_{hair}$ was lower in spring than in summer, while male $CORT_{hair}$ did not differ between seasons (Fig. 5). This model had a greater effect size than
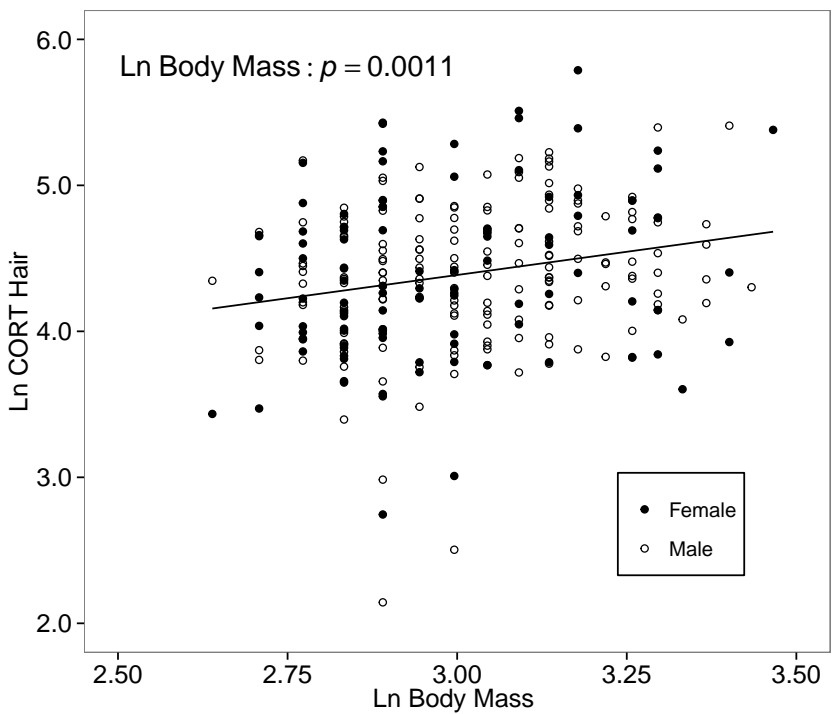

**Figure 4** Hair corticosterone (CORT$_{hair}$; measured in ng/g and ln-transformed) levels increased with body mass in white food mice captured over two years in the Thousand Islands National Park, Canada. The solid line shows a simple regression line; however, the relationship was tested within a linear mixed effects model. Body mass was measured in grams, and CORT$_{hair}$ was measured in ng/g. Both measures were ln-transformed.

**Table 4 Factors predicting fecal corticosterone metabolites of white-footed mice during summer (July–August) in two years, for two separate models (island- mainland comparison, and among-islands comparison).** Linear-mixed effects models were used for analysis (random effect: sampling site, nested within habitat type). Marginal (M) and conditional (C) pseudo $R^2$ ($R^2_{GLMM}$) values are provided. Both models were reduced by removing all non-significant two-way interactions, relative abundance, and ln-transformed body mass.

| Dataset | Fixed effect | $\beta$ | se | df | $t$ | $p$ | $R^2_{GLMM}$ (M), (C) |
|---|---|---|---|---|---|---|---|
| Island and mainland mice | Intercept | 5.695 | 0.78 | 12 | 73.13 | <0.0001 | 0.25, 0.31 |
| | Habitat (Mainland) | 0.022 | 0.12 | 10 | 0.18 | 0.8578 | |
| | Sex (Male) | −0.136 | 0.07 | 201 | −1.98 | **0.0492** | |
| | Year (2016) | 0.620 | 0.07 | 204 | 8.27 | **<0.0001** | |
| Island mice | Intercept | 5.960 | 0.49 | 7 | 12.18 | <0.0001 | 0.21, 0.25 |
| | Sex (Male) | −0.108 | 0.07 | 153 | −1.52 | 0.1320 | |
| | Year (2016) | 0.533 | 0.08 | 154 | 6.52 | **<0.0001** | |
| | Log$_{10}$Isl. Area | −0.002 | 0.03 | 4 | −0.07 | 0.9440 | |
| | Log$_{10}$Isl. Distance | −0.039 | 0.08 | 6 | −0.50 | 0.6320 | |

**Table 5** Factors predicting seasonal variation in hair corticosterone of white-footed mice captured in spring (May-June) and summer (July–August) 2016.

| Fixed effect | $\beta$ | se | df | t | p | $R^2_{GLMM}$ (M), (C) |
|---|---|---|---|---|---|---|
| Intercept | 1.258 | 0.596 | 138 | 2.11 | 0.037 | 0.22, 0.35 |
| Sex (Male) | 0.553 | 0.117 | 136 | 4.73 | **<0.0001** | |
| Season (Summer) | 0.472 | 0.119 | 137 | 3.96 | **0.0001** | |
| Ln Body Mass | 0.891 | 0.186 | 136 | 4.78 | **<0.0001** | |
| Sex × Season | −0.496 | 0.145 | 135 | −3.41 | **0.0009** | |

other models of factors affecting $CORT_{hair}$ ($R^2_{GLMM(M)} = 0.22$, $R^2_{GLMM(M)} = 0.35$; Table 5), indicating a relatively strong effect of season on $CORT_{hair}$ for female white-footed mice.

### Fecal corticosterone metabolites of both sexes were lower in spring than summer

To explore seasonal effects on $CORT_{feces}$ we focused on 2016, for which we had data from both spring and summer ($n = 71$). The final model for comparing $CORT_{feces}$ levels between seasons was reduced from the full model by removing two-way interactions and ln-transformed body mass. Both sexes had lower $CORT_{feces}$ levels in spring than in summer ($p < 0.0001$, Table 6), but there was no difference between sexes ($p = 0.168$; Table 6). The interaction between sex and season, that had influenced $CORT_{hair}$ levels, did not influence $CORT_{feces}$ levels ($t_{63} = -1.423$, $p = 0.160$) so it was dropped from the model.

### Corticosterone in hair and related metabolites in feces were positively correlated

$CORT_{hair}$ values from all collected samples ($n = 333$) ranged from 5.1–398.6 ng/g, with a median level of 79.8 ng/g. Values for $CORT_{feces}$ from all collected samples ($n = 303$) ranged from 30.5–1239.8 ng/g, with a median of 335.4 ng/g. A simple correlational analysis suggests $CORT_{hair}$ and $CORT_{feces}$ of white-footed mice were significantly, although weakly, positively correlated ($r = 0.16$, $t_{178} = 2.196$, $p = 0.015$; Fig. 6).

## DISCUSSION

### White-footed mice do not display island syndrome in the Thousand Islands

Despite the tendency for island wildlife to display morphological and physiological adaptations to insularity (*Matson et al., 2014*; *Holding et al., 2014*; *Spencer et al., 2017*), white-footed mice in the Thousand Islands did not differ in any of these characteristics from their mainland conspecifics. White-footed mice on islands did not display higher relative abundance than mainland mice in disagreement with the general prediction that rodents exhibit particularly high densities on islands (*Adler & Levins, 1994*; *Crespin, Duplantier & Granjon, 2012*; *Cuthbert et al., 2016*), which has been observed for other small vertebrates as well (*Novosolov, Raia & Meiri, 2013*; *Sale & Arnould, 2013*). Additionally, island sites showed a trend toward a greater decrease in relative abundance compared to mainland sites for the two years that we sampled (Fig. 2). This trend may not have

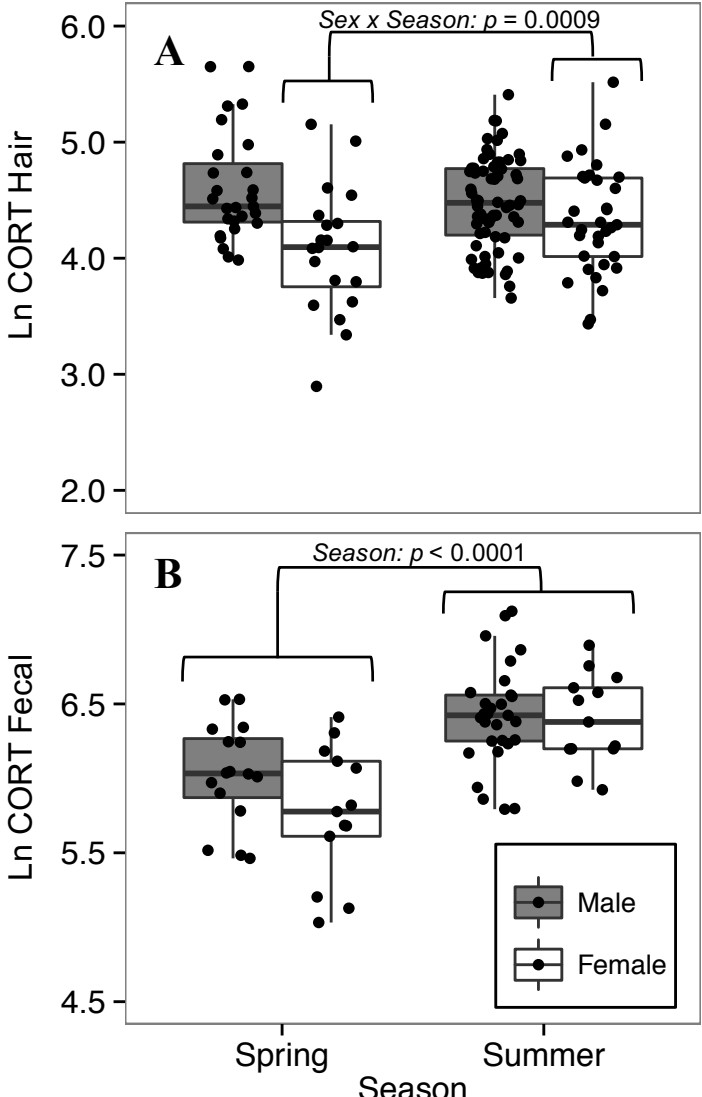

**Figure 5** **Seasonal variation in (A) hair corticosterone (CORT$_{hair}$), and (B) fecal corticosterone metabolites (CORT$_{fecal}$) from white-footed mice captured in spring and summer, 2016.** Females had lower hair corticosterone levels in the spring (May-June) than in the summer (July-August) (sex*season interaction; $p < 0.0009$; A), while both sexes had lower fecal corticosterone metabolites in spring than summer ($p < 0.0001$; B). Both CORT measures were quantified in ng/g and ln-transformed.

resulted in a significant interaction effect between habitat type and year due to the small number of mainland sites that were sampled during both summers. Although this trend only represents two years of data, these results differ from *Adler & Levins (1994)* description of island syndrome, and studies showing that island rodents are more stable, and less prone to drastic changes in abundance than mainland populations (*Gliwicz, 1980*; *Herman & Scott, 1984*; *Tamarin & Sheridan, 1987*). The stability of high population densities on oceanic islands has been partially attributed to marine resource subsidies and climate stability compared to continental systems (*Stapp & Polis, 2003*; *Barrett et al., 2005*;

**Table 6** **Factors predicting seasonal variation in fecal corticosterone metabolites of white-footed mice captured during spring (May–June) and summer (July–August) 2016.** A linear-mixed effects model was used for analysis (random effect: sampling site, nested within habitat type). Marginal (M) and conditional (C) pseudo $R^2$ ($R^2_{\text{GLMM}}$) values are provided. The model was reduced by removing all non-significant two-way interactions and ln-transformed body mass.

| Fixed effect | $\beta$ | se | df | $t$ | $p$ | $R^2_{\text{GLMM}}$ (M), (C) |
|---|---|---|---|---|---|---|
| Intercept | 5.90 | 0.09 | 39 | 64.75 | <0.0001 | 0.30, 0.44 |
| Sex (Male) | 0.21 | 0.08 | 62 | 1.40 | 0.168 | |
| Season (Summer) | 0.45 | 0.09 | 67 | 5.13 | **<0.0001** | |

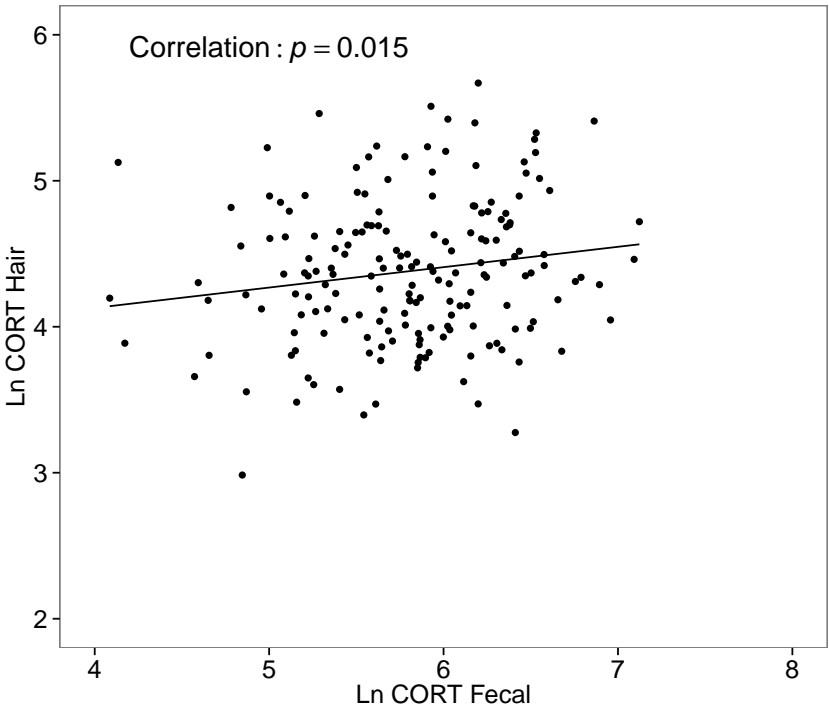

**Figure 6** **Correlation between hair corticosterone levels and fecal corticosterone metabolites for white-footed mice ($n = 180$) captured in the Thousand Islands National Park, Canada.** Both CORT measures were quantified in ng/g and ln-transformed.

*Sale & Arnould, 2013*); factors that do not apply when comparing rodents on the near-shore Thousand Islands to the adjacent mainland.

Small mammals on islands often exhibit large body size (*Lomolino et al., 2012*; *Sale & Arnould, 2013*; *Harper & Rutherford, 2016*); however, we did not detect a difference in body mass between island and mainland white-footed mice in the Thousand Islands. This was surprising, because a high degree of isolation is not necessarily required for demographic or body mass differences among *Peromyscus* populations to occur. *Adler, Wilson & Derosa (1986)* detected a relationship between population density and island isolation of white-footed mice on coastal islands located between 16 and 1,046 m from shore (by comparison, we had large samples of mice from Thwartway and Grenadier islands, which were 2,837

and 1,038 m from the mainland, respectively; Table S1). Deer mice on Anaho Island in Pyramid Lake, Nevada (approximately 1,000 m from the mainland) have significantly greater body length than mainland mice (*Kuhn, Gienger & Tracy, 2016*).

We also found no effect of island area or isolation on body mass of white-footed mice. These negative results regarding patterns between body size and island biogeography do not agree with results for other small mammals in the Thousand Islands. *Lomolino (1984)* found that the body size of meadow voles (*Microtus pennsylvanicus*) and short-tailed shrews (*Blarina brevicauda*) in the Thousand Islands increased as distance from the mainland increased. This pattern was attributed to the ability of larger individuals to cross greater distances on ice during the winter, and subsequent founder effects of large individuals reaching more distant islands (*Lomolino, 1984*). The islands sampled by Lomolino might have been better suited to investigating patterns related to isolation because they were less closely clustered together than those that we sampled. The clustered nature of many of the islands in our study makes their true degree of isolation difficult to determine.

On average, we found that male white-footed mice were heavier than females for both island and mainland habitats. This result is consistent with data from laboratory raised white-footed mice (*Dewsbury et al., 1980*). Greater male body mass has also been found for deer mice collected in the field (*Schulte-Hostedde, Millar & Hickling, 2001*). Male-biased sexual size dimorphism is widespread in mammals (*Isaac, 2005*), and often explained by sexual selection favouring large male body size through competition between males for access to mates (*Trivers, 1972*).

## No difference in CORT levels between island and mainland mice

We predicted that white-footed mice on islands would have lower $CORT_{hair}$ and $CORT_{feces}$ levels than mainland mice, however these predictions were not supported. There was also no effect of island size or distance from the mainland on CORT levels. This result may not be surprising, given that we found no evidence of island syndrome. The lack of an island effect on any of these characteristics in the Thousand Islands suggests either that island white-footed mice experience similar stressors and pressures to mice on the mainland, or that there is a high degree of gene flow between island and mainland white-footed mice. Either of these explanations could be caused by the short distances between islands and from the islands to the mainland, and the freezing of the river in the winter.

The proximity of these islands to the mainland, and to one another, may mean that insular white-footed mice experience similar levels of inter-specific competition to mainland mice. Although the diet of white-footed mice is based primarily on insects, they also forage heavily on seeds (*Manson & Stiles, 1998*). Release from competition with larger granivores, such as squirrel species (Sciuridae), could result in increased body size of *Peromyscus* (*Nupp & Swihart, 1996*). However, we caught red squirrels (*Tamiasciurus hudsonicus*) and flying squirrels (*Glaucomys* spp.) on near-shore islands (Constance and Georgina), and observed gray squirrels (*Sciurus carolinensis*) on more isolated islands (McDonald and Thwartway; N. Stewart, Personal Observation). Although we did not catch eastern chipmunks on any islands, they have previously been caught on islands in

the archipelago (*Werden et al., 2014*). Given the presence of other small mammal species, release from competition might not be a factor on these islands.

The proximity of these islands to shore might cause equal predation risk on the islands and the mainland. Small terrestrial predators, such as weasels (*Mustela* spp.), might occur in low numbers on some of the islands (Grenadier Island; *Werden et al., 2014*). however, avian predators can readily access islands to prey on mice. Coyotes (*Canis latrans*), red foxes (*Vulpes vulpes*), and raccoons (*Procyon lotor*) also inhabit or periodically forage on islands by crossing ice in the winter (*Coleman, 1979*). We experienced somewhat higher numbers of tripped traps on islands than at mainland trapping sites. We visually observed raccoons on a number of the islands, and attribute much of the disturbance to raccoons instead of missed white-footed mice, given that traps were often wide open and moved from their original set point (*Grant, 1976*). Racoons might have been more likely to find traps within the boundaries of islands than mainland expanses, given the small size of most the islands in this study compared to typical raccoon home range size (100–300 ha; (*Kaufmann, 1982*).

High gene flow among islands and the mainland in the Thousand Islands could be attributed to the ability of white-footed mice to swim short distances or cross ice in the winter (*Lomolino, 1989*), and to disperse via transport onboard boats. Despite these dispersal mechanisms, genetic dissimilarity between *Peromyscus* populations can occur at short distances (<500 m from mainland or large island) in other freshwater archipelagos (*Landry & Lapointe, 2001*; *Vucetich et al., 2001*). Genetic studies of white-footed mice in the Thousand Islands would provide greater understanding of the degree of similarity between separate populations in the archipelago.

Studies of the island rule attribute a range of factors to the degree to which body size of small mammals on islands differs from their mainland relative, including island isolation, latitude, and marine subsidies (foraging on aquatic prey; *Lomolino et al., 2012*). Rainfall has also been attributed to increased body size of small mammals, and used as evidence of the island rule being more of a "resource rule" (*McNab, 2010*). It is because of this range of explanations for observed island-mainland differences that we were drawn to study island rodents in a near-shore archipelago, where climatic and ecological variation is limited between island and mainland sites. However, based on our results, a significant degree of ecological isolation may be required for island syndrome to be detectable across an archipelago.

## Hair corticosterone increased with body mass

Body mass was a positive predictor of $CORT_{hair}$ in white-footed mice, but not $CORT_{feces}$, which suggests this might be a hair-specific CORT observation, as opposed to representative of baseline CORT levels. Because body mass is positively correlated with age in *Peromyscus* (*Chappell, 2003*), the positive relationship between $CORT_{hair}$ and body mass might be indicative of a relationship between mouse age, moulting, and $CORT_{hair}$ levels. Moulting in *Peromyscus* occurs before or following energetically demanding time periods, such as breeding (*Pierce & Vogt, 1993*), although some hair replacement likely occurs year-round outside of complete moults as in deer mice (*Tabacaru, Millar & Longstaffe, 2011*). Moulting

is in part regulated by CORT, because steroid hormones have an inhibitory effect on moulting in *Peromyscus* (*Garwood & Rose, 1995*). As a result, hairs grown during complete moults might have relatively lower CORT concentrations than replacement hairs grown following a complete moult. This would result in heavier individuals (which are likely older and have increased in mass since their last moult) having higher $CORT_{hair}$ when compared with younger mice that have more recently grown their adult pelage. Testing this relationship would require more specific knowledge concerning the exact age of each mouse, and time since their last moults.

In American pika (*Ochotona princeps*), hair CORT was strongly influenced by body size (measured by cranial diameter; *Waterhouse et al., 2017*), but in the opposite direction compared to white-footed mice. Larger American pikas had lower hair CORT, which the authors attributed to the negative relationship between mass-specific metabolic rate and GCs (*Haase, Long & Gillooly, 2016*). The conflicting directionalities of relationships between hair CORT and body size for small mammals demonstrate the need for more studies concerning internal factors affecting hair GCs.

Hair GC levels are an integrative measure of HPA activity, because they will reflect both an individual's phenotype related to their baseline GC levels (*Fairbanks et al., 2011*), but can also be influenced by an animal's exposure to stressors (*Bryan et al., 2015*; *Scorrano et al., 2015*). Because the two measures are representative of different time frames, it might not be surprising that CORT in hair and its related metabolites in feces are not correlated with the same measures (body mass and condition), and demonstrate different seasonal patterns. We did find, however, that $CORT_{hair}$ and $CORT_{feces}$ were positively correlated for white-footed mice. Such correlations presumably represent the influence of individuals' baseline GC levels on both $CORT_{hair}$ and $CORT_{feces}$, however, the relatively weak correlation may reflect matrix-specific time-frames of GC secretion.

## Sex-specific seasonal variation in corticosterone

Seasonal differences in GC levels are common among vertebrates (*Romero, 2002*). $CORT_{feces}$ levels of white-footed mice of both sexes were higher in the summer than the spring (Table 6, Fig. 5B), in agreement with previous studies of *Peromyscus* (*Harper & Austad, 2000*; *Harper & Austad, 2001*). High summer $CORT_{feces}$ levels could be attributed to increased GC levels associated with reproduction (*Harper & Austad, 2001*); however, breeding occurs in white-footed mice during early spring and lasts throughout the summer months (*Pierce & Vogt, 1993*). Alternatively, high summer $CORT_{feces}$ levels could be caused by increased abundance of white-footed mice compared to the spring (*Hayssen, Harper & DeFina, 2002*), which is consistent with our finding that abundance was higher in the summer than the spring in the Thousand Islands. Although we did not find a relationship between CORT and relative abundance across sites, CORT and population density may be correlated within areas as densities change during annual cycles.

In contrast to data for $CORT_{feces}$, only female white-footed mice showed seasonal variation in $CORT_{hair}$, increasing from spring to summer (Table 5, Fig. 5A). Male $CORT_{hair}$ did not differ between seasons. The contrasting sex-specific results between $CORT_{feces}$ and $CORT_{hair}$ patterns might be the result of the two measures differing in their

representative time-scales of CORT secretion (*Mastromonaco et al., 2014*). Similar to our proposed explanation of the relationship between $CORT_{hair}$ and body mass, it is important to consider the effect that CORT has on hair growth. Female deer mice have been observed developing seasonal moults during pregnancy; however, moulting ceased following the birth of their young (*Collins, 1923*). If hair growth in adult female white-footed mice occurs during pregnancy (similar to deer mice), our observation of low $CORT_{hair}$ in females may represent low CORT levels during spring pregnancies. Low CORT during pregnancy occurs in laboratory rodents and it has been suggested that attenuation of the HPA axis during pregnancy protects fetuses from adverse effects of high GCs (*Reeder & Kramer, 2005*; *Brunton, Russel & Douglas, 2008*). Although we did not assess the influence of reproductive condition on $CORT_{hair}$, a study focused on the relationship between reproductive condition and $CORT_{hair}$ in wild rodents would be useful.

Sex differences in GC levels between seasons occur in other wild rodents and are attributed to interactions between GCs and sex hormones, and differences in parental behaviour (*Romero et al., 2007*; *Schradin, 2008*; *Bauer et al., 2014*). In addition to reproductive influences on $CORT_{hair}$, higher female $CORT_{hair}$ levels could be attributed to increased aggression among territorial females when population density increases during the summer (*Wolff, 1993*). The significant interaction that we observed between sex and season on $CORT_{hair}$ raises the question of why males did not differ as much as females. The contrasting results between sexes shown for $CORT_{feces}$ and $CORT_{hair}$ emphasize the importance of considering the timeline represented by the sample material.

## CONCLUSIONS

The two aims of our study were: 1. to test if white-footed mice in the Thousand Islands display characteristics of island syndrome, including greater body mass and higher relative abundance than their mainland counterparts, and 2. to test if there were also differences in stress physiology between island and mainland mice. Local populations of white-footed mice in the Thousand Islands did not differ systematically in their abundance, body mass, or hair and fecal GC levels, compared with white-footed mice on the nearby mainland. We suggest this may be due to the relatively short distances between the islands and the mainland, the clustered nature of the islands, and that the St. Lawrence River freezes during the winter allowing for possible movement of mice and predators between islands and the mainland. Because we found no island-mainland differences in either body mass or GCs, our results leave open the possibility that on more isolated islands, where the community structure is distinctly different from mainland habitats, decreased interspecific competition and predation may cause changes in the stress physiology of rodents. However, such studies would need to account for the many variables (temperature, precipitation, and day length) that differ between more distantly-spaced island and mainland habitats. Although our initial predictions were not supported, our incidental findings of a relationship between body mass and $CORT_{hair}$, and that measures of stress differed with sex, season, and matrix (hair or feces) emphasizes the complexity of inferring physiological state from hormonal profiles.

## ACKNOWLEDGEMENTS

We thank Chantelle Penny, Tess Ward and Don Stewart for assistance with field work, Christine Gillman and Stephanie Matteer for assistance with hormone analysis, Gabriel Huebsch for guidance with spatial analysis and map-making, and Aaron Shafer, Jeff Bowman, Jim Schaefer, and three anonymous referees for comments on previous drafts of the manuscript.

### Funding

This work was funded by the Natural Sciences and Engineering Research Council, Canada (No. RGPIN-04158-2014), and the Toronto Zoo. The funders had no role in study design, data collection and analysis, decision to publish, or preparation of the manuscript.

### Grant Disclosures

The following grant information was disclosed by the authors:
Natural Sciences and Engineering Research Council, Canada: RGPIN-04158-2014.
Toronto Zoo.

### Competing Interests

The authors declare there are no competing interests.

### Author Contributions

- Nathan D. Stewart conceived and designed the experiments, performed the experiments, analyzed the data, prepared figures and/or tables, authored or reviewed drafts of the paper, and approved the final draft.
- Gabriela F. Mastromonaco analyzed the data, authored or reviewed drafts of the paper, and approved the final draft.
- Gary Burness conceived and designed the experiments, authored or reviewed drafts of the paper, and approved the final draft.

### Animal Ethics

The following information was supplied relating to ethical approvals (i.e., approving body and any reference numbers):

The Trent University Animal Care Committee, in accordance with the Canadian Council on Animal Care (CCAC), provided full approval for this research (Trent University Animal Use Protocols No. 23877 and No. 24341).

### Field Study Permissions

The following information was supplied relating to field study approvals (i.e., approving body and any reference numbers):

Field experiments in the Thousand Islands National Park were approved by Parks Canada (Permit No. 22959).
## Data Availability

Raw data are provided as Supplemental Material.

## Supplemental Information

Supplemental information for this article can be found online at http://dx.doi.org/10.7717/peerj.8590#supplemental-information.

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
