# Peer review of "No island-effect on glucocorticoid levels for a rodent from a near-shore archipelago"

_PeerJ, doi:10.7717/peerj.8590_

## Round 0.1 · original submission · Major Revisions

We have received three detailed reviews for your manuscript. All reviewers concurred that it is an interesting study addressing an important topic in ecology. However, they also raised several important issues regarding the justification of the predictions, the study design and statistical analyses that warrant extensive revisions.

In particular, the justification for choosing the sampling sites and trapping procedures (timing of captures at different sites, tripped traps for examples) were identified as being problematic by reviewers. One of the reviewer suggested that your study setting did not met the conditions for the population- and individual-level adaptations to evolve and thus for mainland and island differences to be detected. You will obviously need to answer this criticism convincingly. More details concerning the sample sizes also need to be provided. Statistical model parametrization and model selection were deemed confusing and need thorough clarifications to assess their validity. A report for the validation of measure hair and fecal corticosterone levels should also be included. Finally, the reviewers provided several other comments that need to be integrated in the next version of the manuscript.

Reviewer 1 ·

Basic reporting

Very well writing and concise. In some areas the references and context could be expanded and I give specific comments below in the General Comments section. The figures and tables look good but could use some additional details about statistical differences to aid readers. Raw data are shared. More comments in the General Comments section.

Experimental design

Fits Aims and Scopes of the journal and is original research. The research question is defined but predictions need some elaboration (I detail this in the General Comments below). The authors nicely describe the knowledge gap about "island syndromes". The Methods need more details and I make specific suggestions in the General Comments below.

Validity of the findings

There are some issues with the statistical approaches and more details are needed to evaluate these. I discuss this and provide suggestions below.

Additional comments

This manuscript examines aspects of the “island syndrome” in wild deer mice in islands and mainland areas in Canada. The island syndrome is characterized as the constellation of phenotypic traits that differ between individuals living on islands versus those conspecifics on the mainland. For example, the authors describe how rodents typically show increased body size/mass on islands vs. mainland areas and, as the authors indicate, this may be because of differences in the abundance of food, interspecific competitors, or predators

The authors quantified body mass of mice on islands and mainland areas as well as two measures of their stress physiology (hair corticosterone and fecal corticosterone levels). The authors indicate that differences in stress physiology between island and mainland populations have not been investigated in detail. Mice were captured on different islands (n=11) and mainland areas (n=5) over a few months in the spring and/or summer in two consecutive years. Hair corticosterone and fecal corticosterone levels were measured from samples collected during trapping and body mass was recorded. The density of mice was also estimated.

The authors show that mice captured on islands and those on mainland areas did not differ in body mass, fecal corticosterone, or hair corticosterone. They also did not find a difference in mouse density on islands vs. mainland areas. However, they do show seasonal shifts in fecal and hair corticosterone levels and that heavier mice had higher hair corticosterone levels. They also report a weak but significant positive association between fecal corticosterone and hair corticosterone.

I enjoyed reading this manuscript and it addresses an interesting question about physiological differences between island and mainland populations that has been rarely addressed and uses a comprehensive approach to do so. The fieldwork undertaken is non-trivial and it was impressive to see that the authors had multiple replicates of the island and mainland populations and did the study across two different years. Although the predictions were not supported, the manuscript is still a valuable contribution in my opinion.

I have a few major and minor comments for the authors below.


Major comments:

I have a few requests for clarifications regarding the statistical methods. First, in the island-mainland comparisons, why was island included as a random effect nested within habitat type? Shouldn’t habitat type be a fixed effect in these comparisons? Second, the rationale for the a priori interactions being included needs to be described. Currently (line 193), it is not adequately described and it seems unusual that preliminary analysis was used to assess which interactions were included. Adding references for why interactions were included for the statement “based upon a priori knowledge” (line 193) also seems needed. Third, the procedures to select the ‘top’ model is not very clear or transparent and needs to be described in greater detail (lines 196-200). Why not use model selection under this approach? Finally, whether or not the fixed effects in the models were or were not standardized should be reported here.

I find the basis for the predictions presented in the Introduction to be slightly unclear (lines 88-95). Specifically, why exactly would you predict that if mice on islands were more abundant a and larger than mainland mice that they would have lower hair and fecal corticosterone levels? That seems quite divergent from what is presented above (and the literature as a whole) where animals in poor condition (such as lower body mass) or those experiencing higher densities would have higher stress measures.

As a whole, I find that the Methods could use more details in certain areas and provide a number of specific comments for where to add detail in the minor comments shown below. More space for Methods is available as it is noticeable that the Methods (lines 97-211) are substantially shorter than the Discussion (lines 302-446).

It isn’t clear to me why trapping occurred at very different time periods between the two years. Capture of mice in 2015 (July-August) and 2016 (May-June) was very different and it would be helpful for the authors to indicate why this occurred because seasonal changes in reproductive activity or within-season changes in density could affect the results. How were the effects of reproductive condition on hair or fecal corticosterone levels controlled for here? Additionally, how synchronous were the sampling of the mainland and island areas? I’m just wondering if the negative results might be due to variation in fecal or hair corticosterone levels due to differences in the seasonal time of sampling.

Whether or not the assays to measure hair and fecal corticosterone levels were validated should be reported in the Methods. As indicated in my minor comments below, it would also be useful to provide more details on the hormone analysis methods.



Other minor comments:

As a general comment, the introduction is nice and concise but in some areas more details would be helpful. For example, when you write “morphological and behavioural traits associated with island syndrome… (line 44), could you indicate what specific traits you are referring to here? What have these studies found?

Additionally, the Introduction has a sharp transition to the stress physiology component and how island syndrome and stress physiology are linked should be made clear before this paragraph.

Line 52 – it seems odd that the references for the effects of density on stress levels are from a primate species and birds when this is a topic that has been studied at length in rodents including mice (reviewed in Creel et al. 2013 Functional Ecology).

Line 56 – it seems like there should be a caveat here for elevated GCs “can result” in negative health and fitness implications? This McEwen and Wingfield paper is arguably from the perspective of lab animals and humans. Throughout the rest of the manuscript, the authors have done a great job at referencing primary research articles (not reviews) but here they should continue that trend and give some examples where long-term elevations in GCs resulted in a reduction in health and fitness in a wild animal.


Line 64- whether its 2 or 72 hours, this is a long time difference and is highly dependent upon the species.

Line 86 – what is the basis for the statement that corticosterone is the major GC in mice and rats? Is there a reference for this?

Line 113-114 – because of my other comment above about the timing of captures and how it might vary between island mainland sites, it would be useful to provide details on when trapping occurred and at each site. How much synchrony occurred between trapping at the island and mainland sites?

Line 121 – I would think that estimates of density would only be accurate with at least 3 nights of trapping. Could you show some more details about how much trapping effort existed between island and mainland sites and if it differed? Did trapping effort impact density estimates?

Line 135 – this level of detail is nice but could the rationale for why this variation in coat colour might impact the hair stress levels be reported here?

Line 138 – is that 6 hour period from collection to freezing likely to impact the fecal stress levels? How do you know if the samples were collected at ~1800 h (when traps were set) or 645 h the following day (when traps were checked)?

Line 141- does time in the freezer impact fecal stress levels?

Line 146 – could you report here the trap nights at the different sites and if it differed among the island and mainland sites?

Line 154 – the basic details of the extraction procedure should be provided here instead of just the two references.

Line 155 – does time spent in the freezer impact the hair corticosterone levels?

Line 167 – are these CVs for hair or fecal corticosterone assays? Can you report how many assays in total were run.

Line 183 – it would be very useful to describe how reproductive condition was assessed and it how might impact the measures of fecal or hair corticosterone. How was pregnancy status in females assessed?

Line 211 – the version number of all the R packages used here should be indicated.

Line 255 – this “condition-dependence” where hair stress levels and body mass are positively correlated would negate one of the statements in the Introduction that high stress levels negatively influences health and fitness.

Tables – I suggest reporting exact P-values unless they are <0.0001

Figures – thank you for reporting the raw data in these figures but why not show the statistical differences (e.g., seasonal effects) or correlations among some of the variables? It makes it challenging for readers that only look at figures to discern if these are statistically significant or not.

Reviewer 2 ·

Basic reporting

The manuscript ”No island-effect on stress for a rodent from a near-shore archipelago” assesses whether near-shore island populations of white-footed mice show population-level (density) or morphological and physiological indications of “island syndrome”. Island syndrome is a group of systematic differences in characteristics that tend to be associated with insular living, including higher population densities, larger body size, longer longevity, lower reproductive output, and lower aggression towards counterparts. Three main factors have been highlighted as essential for the evolution of island syndrome: reduced local interspecific competition thus allowing for increases in population densities, isolation from other suitable habitats thus reducing or eliminating dispersal opportunities so that high densities are maintained, and low density-depressing forces such as lower predation pressure, which along with high population densities, tends to select for larger body size, lower sexual maturation rate, and lower reproductive rates, as well greater tolerance of conspecifics and lower aggression levels.
While many differences in morphological and life-history traits between mainland and island populations are well described in the literature, physiological differences have been less explored. Here, the authors investigate whether insular living is also associated with consistent differences in HPA regulation. They do this by assessing long-term corticosterone deposition levels in fur samples, and short-term corticosterone secretion via faecal metabolite concentrations, in captured white-footed mice from 11 islands and 5 mainland sites. Lower predation pressure and lower reproductive rates may be associated with lower GC levels for instance, while higher population densities might be expected to associate with higher GC levels. How these population- and individual-level differences, that together lead to the island syndrome, balance each other out to affect individual GC levels is still not fully understood and is very much worth investigating.

The authors tackle three main subjects in this manuscript:
1) assess whether island populations are found at consistently higher densities and are morphologically and physiologically different from mainland populations;
2) investigate whether island syndrome characteristics are influenced by island size and distance from mainland, and
3) explore associations between hair and faecal CORT with individual characteristics (sex and size), habitat and seasonality.

MAIN COMMENTS:
This study is generally well written and referenced, although it lacks important information at times (see below).

The research questions are well defined, and they are certainly interesting. However, I fear that this study has a major flaw that is hard to get around. The authors acknowledge that the islands are neither isolated (mice may swim between the islands in the summer and walk over the ice between them in the winter) nor entirely free from terrestrial predators, who are capable of moving between mainland and the islands, especially in the winter, and some small populations of predators may be established in some of the chosen islands. This study seems to be a good study, the research question is interesting, and the study seems to have been carried out correctly with a good sampling regime and decent samples sizes. Why did the authors choose this set of islands to test these questions? The choice of location appears to me to make it impossible for the authors to test the questions they set out from the start. Without isolation and reduced mortality pressure (in this case from predation), it stands to reason that white-footed mice would not show any population-level or morphological adaptations to island life as expected according to the island syndrome. Without these, there must be very little reason to expect the same populations to nevertheless show differences in glucocorticoid levels.
The authors point out that island syndrome has been detected in other rodent species at the same national park, although the studies were performed on different islands that were less clustered and thus more (truly) isolated. This point highlights how isolation may not mean the same thing when referring to dispersal opportunities, which directly affects population densities and have knock on effects on body size, behaviour and reproductive rates, and when referring to isolation from predators, which tends to select for smaller body sizes. Dispersal opportunities are likely affected by nearby islands and so potentially distance to nearest land mass rather than distance to mainland should be estimated. Adler & Levins 1994 provide very good explanations and details regarding the importance of sinks in determining morphological responses to island-living. As for predation pressure, I agree that distance to mainland may be the best estimate, at least with regards to terrestrial, larger predators.
Another important point to clarify is that the explanation provided for the species in which island syndrome characteristics have been detected (meadow voles and short-tailed shrews, Lomolino 1984) was that larger individuals were more likely to walk longer distances on ice and found populations at islands that would have average larger body mass. This reasoning is very different from the usual assumption that larger body sizes are an adaptation to the local conditions, such as low predation rates and higher population densities at said islands. As the potential mechanism responsible for the larger sizes of individuals in these species on some of the islands is different from that assumed to bring about larger body sizes and other island syndrome characteristics, I’m not sure one should consider those results as evidence of island syndrome in rodents at the Thousand Islands National Park. However, regardless of the means by which these species attain larger body sizes at the studied islands, I would like the authors to provide a bit more information on whether there was any overlap in the chosen islands in both studies, why they did not choose the same islands and why they did not work on those species for which body size differences had already been documented, as a start point to explore physiological differences between mainland and insular individuals.
Similarly, the authors did not find significant effects of island size on island syndrome characteristics. As islands need to be considerably large for their size to have noticeable effects on population densities, it is likely that the range of sizes of the islands chosen for this study was not broad enough to detect such differences.


MINOR COMMENTS:
Figure 3: Since year does not significantly affect body mass (as main effect or in interaction with habitat) and in itself (2015 v 2016) is not a main focus on the study, I would merge the data and combine the two years, presenting only sex and habitat. The same goes for hair CORT results, and although fecal CORT levels are higher in 2016 than in 2015, unless you have a particular hypothesis or explanation as to why this might be, I do not think this is your most interesting result and that it needs to be visualised. Sample sizes for mainland females seem quite low.

Experimental design

EXPERIMENTAL DESIGN:
Be very transparent with your sample sizes for each response variable. Hair CORT analyses have smaller sample sizes relative to body mass analyses. Is this discrepancy due to the removal of all samples that had > 10% CV? How many samples were omitted in total? How many used in the analyses?
I would like to see more information on how the islands were selected as well as their distance to their nearest land mass, not just mainland. It seems to me that Mermaid, Beaurivage and Camelot Islands add very little to the overall dataset. In addition, Mermaid Island uniquely had its traps placed along transects and Camelot Island artificially appears to be the most isolated island (from distance to mainland) when it is actually very close to multiple other islands (Figure 1).

Methods:
The paragraph starting on line 144 (relative abundance) requires more explaining. For instance, please provide the correction factor equation you used to control for tripped traps. Also, what is your interpretation of the tripped traps occurrences? Would they have been tripped by white-footed mice or by other species? None of this is explained and it is not obvious to the regular reader.

Validity of the findings

Results:
I am not sure I fully understand how you analysed your data, or at least how you have chosen to present it. For instance, does Table 1 present the results of 2 models or multiple models within each dataset. It looks like you present the results from 1 final model with all resulting significant effects, however, you state that the effect of year was tested using summer data from 2015 and 2016, while you test the effect of season using 2016 spring and summer data. Each of these analyses requires a different subset of your data, and therefore could not have been done in just one model. So either the statistics are not presented correctly, or they are not described correctly (paragraph starting on line 225).

Paragraph starting on line 279: As the season by sex interaction is significant, you should not report on the main effect of sex. The two factors are not independent.

Discussion:
Paragraph starting on line 317: the authors should clarify what they mean by “a high degree of isolation”, explain whether isolation was measured in similar ways and how it compares between the studies cited and the current one. How low a degree of isolation is required for population differences to be detected?

Line 333: The authors found that males were heavier than females independently of where they were sampled (mainland or islands). As is, it is unclear whether “in the Thousand Islands” relates to the whole national park including mainland areas or refers to the sampled islands only. If the latter, then the statement is not fully correct: yes, males on the islands are heavier than females, but so are males at the mainland sites.

Hair corticosterone increased with body mass: The first paragraph in this section needs some clarifications.
Do you mean?:
1 – older individuals are heavier
2 – heavier individuals are more likely to breed
3 – breeding is stressful resulting in higher corticosterone levels
4 – corticosterone inhibits moulting
5 – resulting in breeding individuals having had their current coat for longer and put on weight since their last moult (though not clear why this is relevant in this context – line 389)
6 –incomplete moults do not replace all fur, thus older hairs store higher CORT
7 – concluding that heavier individuals are more stressed due to breeding which also inhibits moulting resulting in incomplete moults which means that individuals have had their hair for a longer period of time allowing for more CORT to be deposited in the hairs.
8 – and in parallel: younger individuals, which would be smaller, “have more recently grown their adult pelage” = lighter individuals may have moulted more recently and thus have lower CORT levels in their fur?
If I did understand all these arguments correctly, then they need to be explained better in that first paragraph.

Line 403: The correlation between CORT hair and CORT faeces, as reported by the authors, cannot be “similar” as the one reported for eastern chipmunks, because the latter is not significant and thus the two CORT measurements are not correlated.

I find the interaction between sex and season in CORT hair levels very interesting. The authors have, understandably, focused on the females, as this is the sex that differs between seasons. However, it takes both sexes to make the interaction significant. I would like to know the authors’ interpretation for males maintaining high CORT levels in both Spring and Summer. What is it about the mating or social system of this species, that the authors argue that females can have reduced CORT levels in the Spring, despite being pregnant, yet males’ levels remain high? A little more information regarding the species’ reproduction (number of litters, number of young per litter, etc) in the methods section would be quite useful.

Additional comments

This study clearly has value but, as is written, I think it highlights what I see as a fundamental flaw in it – that you could not possibly test your main questions in the set up (Thousand Islands National Park) that you used because the location does not have the basic requirements/conditions for the island syndrome to occur. Your results relating to island syndrome thus cannot provide any evidence for or against your main hypotheses. I wonder whether you could instead, focus on the link you found between CORT hair and CORT faeces with seasonality and sex, and having established that there are no differences between the populations, use that dataset to explore further questions. For instance, you discuss that both matrices reflect basal metabolic rates, but faecal CORT is additionally more strongly influenced by short-term stressors. Would you expect the measurements for females in the Spring, when faecal CORT levels are lower to correlate more strongly with hair CORT?

Reviewer 3 ·

Basic reporting

Overall the paper was strong in this respect. I have some minor comments to improve the figures and tables (see attached comments).

Experimental design

The field methods were appropriate. I do however have substantive questions and potential concerns with the statistical analyses. These concerns (and suggested improvements and points of clarification) are detailed in the attached comments.

Validity of the findings

Assuming that questions regarding statistical approach (detailed in attachment) are addressed, the findings are interpreted appropriately. I have some minor feedback (detailed in comments) where additional information about the system will facilitate reader understanding of data interpretation.

Annotated reviews are not available for download in order to protect the identity of reviewers who chose to remain anonymous.

---

## Round 0.2 · Major Revisions

We have received two additional reviews from the same reviewers who previously assessed your manuscript. Both reviewers noted that the revisions improved your work. Unfortunately, they also both identified problems remaining in the current version. I thus give you an additional opportunity to revise your manuscript, to respond to the comments provided.

In particular, the validation of the measures of stress hormones in faeces and hair needs to be addressed in details. Additional revisions to the modelling section and abundance analysis are also suggested. Some of the figures also need to be improved.

Reviewer 1 ·

Basic reporting

The manuscript is very well written and clear. All of the revisions help to guide the reader - thank you for these.

The figures and tables are all appropriate but the one issue that I wanted to mention is that Figure 4 has a regression line that doesn't match up with with the data (it's origin goes beyond the lowest value of body mass observed). Similarly, Figure 6, the regression line goes beyond the maximum value of fecal CORT observed.

Experimental design

In the original review, I asked the reviewers to provide more details about their validations of hair and fecal corticosterone levels and asked to provide more details on the hormone analysis methods. I don't really think that their response (adding two citations) is adequate and we still need more details. For example, what does it mean that the corticosterone antibody used in this study was "similar" to one used previously? I think that this is a very important point and needs more detail for the authors to justify their use of the antibodies used for faeces and hair.

The authors indicate in their response letter that "During the 6-hour period from collection to freezing, the samples were kept on ice packs, which slows down any degradation of the hormones in the feces until the samples can be frozen" but provide no references for this statement. I would like the authors to justify this statement and add some sentence to the main text of the manuscript. Similarly, the authors indicate in the response letter that "Long-term storage in a –20C freezer can cause some degradation of glucocorticoids in the feces. However, tests in our lab on the re-extraction of fecal samples over time did not show any significant changes in fecal glucocorticoid metabolites within one year at –20C." but where are these analyses? Shouldn't they be provided for us to judge ourselves?

Validity of the findings

These all seem appropriate and thank you for the clarifications to the statistical analyses.

Additional comments

Thank you for making all of these detailed revisions to this manuscript. In light of the recent MacDougall-Shackleton et al. (2019 Integrative and Comparative Biology) paper about how "stress" is not the same as "glucocorticoids" and here you are only measuring glucocorticoids, might it be worth to revisit the title and use "glucocorticoids"?

In the original submission, I thanked the authors for providing citations to the original research (rather than reviews) and I appreciate that they have continued this approach in the revised manuscript. However, in my original review, I asked the authors to provide some references that show how elevations of GCs in wildlife result in a reduction in health and fitness. They now provide two citations (Wright et al., 2007; Hing et al., 2016) but after looking at these, I'm not really sure these meet that aim and they are reviews. E.g., the Wright et al. just reviews how anthropogenic noise may induce physiological stress. Hing et al. is another review that focuses on One Health and doesn't necessarily address this point. Are there better references here to justify this point. Note, DOI for the Hing et al. citation that was added doesn't work.

Reviewer 3 ·

Basic reporting

In general, basic reporting is done to a high standard although I have some smaller points that I address in specific comments to the authors below. Revisions have addressed reviewers’ requests for additional background information in specific areas to aid in understanding the context of the study and interpretation of the results. Figures are clear (except see questions about figure 2) and raw data is shared.

Experimental design

Revisions have been made that address concerns raised by original reviewers about experimental design and methodology. These changes explain why methodological choices were made based on precedent in the literature and logistical constraint (e.g. why these islands were chosen for the study when findings reveal they don’t appear to exhibit island syndrome) and provide more detail about how certain procedures were done (e.g. assay methodology, statistical analyses). While I still find some parts of the reports on statistical methods to be difficult to follow at times, I find this section improved over the previous draft and have provided some additional suggestions on wording that may clarify lingering questions

Validity of the findings

Revisions have made interpretation of findings more clear. I still have one comment on statements comparing changes in abundance over time between mainland and island sites which I highlight below.

Additional comments

Line 64: I find the term matrices surprising in this context. Perhaps just “materials.”

Line 136: I am assuming that females can rear multiple litters over the course of the summer? Specifying this may aid in interpretation of female GC levels.

Line 148: Consistency in what?

Lines 232- 260. I appreciate the revisions that you have done to clarify your models- this section is easier to follow. However, I still get lost in some of the details and can’t keep track of what terms go in what model easily. I think it would be much easier if for each of the 3 general model types you identify (line 235) you just spell out exactly what terms were used in the full model. I imagine this looking something like “We tested our outcomes of CPUE, GCs … in three different model types: A,B, and C. For analyses testing differences between mainland/ island sites, full models included random effect A and fixed effects for X, Y,Z and Y *X.” It might take a little more space, but I think it will help your readers understand exactly what is happening. Then you can just report in your results section “in our final reduced model X significantly predicted abundance (describe direction here). Y, Z, and their interaction were not significant in the initial full models and were excluded from the final reduced model.” This way you’re spared reporting what got dropped from what subset of models in your methods section, which I think can get confusing. You seem to have done this in some sections of the results.

Section starting at Line 328 and Line 349: While I think it is good to spell out what terms are in and out of your final models, this section gets a little bit confusing because it reports results, but doesn’t spell out for every result whether the p-values come from full or reduced models. This is especially confusing because presumably the p-values for non-significant terms shifted a little every time you did a single deletion of one variable and re-ran the model, so the reader can’t tell whether the p-values you’re reporting for non-significant terms come from a maximal model, or a partially reduced but not final model. To me it makes sense to only report p-values for final reduced models, but to also say that variables x, y, and were non-significant in full models and were deleted from the final reduced model.

Line 373-Line 374: This figure shows a correlation in trapping of sites between the two years. It is not clear how this figure shows a decrease in abundance that is greater in island than mainland sites. To do so, we’d need to have a best fit line showing the correlation between island sites and a second line show the correlation in trapping results between mainland sites so that there was some way to see a difference between the two. This could also be enhanced by a caption indicating that results above the 1:1 line show increase in abundance and results below the 1:1 line show a decrease in abundance across time. However, I'm not convinced that this is the best way to show the results--- you could also report in a bar graph. More importantly, in your results for your abundance analysis (Table 1), you do not report a significant interaction between habitat type and year, so I don’t understand how you have the support to make this claim that there is a greater decrease in abundance in one habitat type than the other.

Line 440: So implicit in this is the idea that raccoons are more likely to trip traps on island sites than on mainland sites. Do you know anything about raccoons on the mainland- do think it’s fair to say they may be at lower density than on islands or have different behavioral patterns?

Line 457: Word missing.

Line 470: Do you know anything about hair turn-over outside of molt? Is it conceivable that animals shed so much hair between molts that this could be the case? Part of assessing the likelihood of this may be knowing if animals molt multiple times a year or just one time a year.

Figure 2: This figure should have an R2 value on it? Also see previous comment on interpretation of this figure in the discussion.

---

## Round 0.3 · accepted · Accept

The authors did a great job when addressing the previous comments provided by the reviewers. As a result, the manuscript is now much improved and I am thus pleased to recommend its acceptance.